# Potential of the Geometric Layer in Urban Digital Twins

Andreas Scalas * , Daniela Cabiddu , Michela Mortara and Michela Spagnuolo

Istituto di Matematica Applicata e Tecnologie Informatiche "Enrico Magenes", Consiglio Nazionale delle Ricerche, Via de Marini 6, 16165 Genova, Italy; daniela.cabiddu@cnr.it (D.C.); michela.mortara@cnr.it (M.M.); michela.spagnuolo@cnr.it (M.S.)
* Correspondence: andreas.scalas@ge.imati.cnr.it

**Abstract:** A urban digital twin is the virtual representation of real assets, processes, systems and subsystems of a city. It uses and integrates heterogeneous data to learn and evolve with the physical city, providing support to monitor the current status and predict/anticipate possible future scenarios. In this paper, we focus on the issues and potential related to the geometric layer of the city digital twin. On the one hand, detailed 3D data to reconstruct the urban morphology very accurately might not be available, and planning a new survey is costly in terms of money and time. On the other hand, the more the geometry adheres to the real counterpart, the more accurate measures and simulations related to the urban space will be. We describe our approach to develop the geometric layer of the digital twin of the city of Matera, in Italy, using only pre-existing public data. Specifically, our method exploits available digital elevation models from a previous regional aerial survey and integrates them with data coming from OpenStreetMap to generate an as-precise-as-possible 3D model, annotated with heterogeneous semantic information. We demonstrate the potential of the geometric layer by developing two geometric characterisation services, namely route slope extraction and light/shadow maps according to a specific date and time. In the next steps, the computed attributes will help to answer specific objectives which could be of interest for the Municipality, such as personalised optimal routes taking into account user preferences including slope and perceived environmental comfort.

**Keywords:** 3D modelling; urban intelligence; city digital twin

## 1. Introduction

The concerns for a sustainable future are tightly connected with the spread and continuous growth of urban areas, and how these will adequately address fundamental issues such as healthy life and well-being, sustainability, resilience to climate change, etc. Understanding the connections and integrating urban planning with new enabling technologies (i.e., modelling and simulations, Internet of Things, Artificial Intelligence) is becoming the new paradigm of any new approach to the prediction of future scenarios of urban sustainability and to the generation of long-term policies for the future of sustainable cities. Urban Intelligence (UI) is an eco-system of infrastructures and services that allows the creation of a Digital Twin (DT) of a complex real/physical system such as a city and its various systems and subsystems (e.g., transportation, energy distribution, water usage, population, education, health, cultural heritage, etc.) [1,2].

Urban DTs can be seen as an evolution of the Smart City concept, developing new directions: (1) integrated and intelligent systems for the government of the city, using and integrating heterogeneous data (from sensors to Citizen involvement) controlled by multidisciplinary optimization approaches; (2) a flexible and adaptive digital model that learns from and evolves with the real city; (3) a predictive model capable of anticipating future scenarios [1].

A number of urban DTs have been implemented in India, Southeast Asia, and Europe [3–6]; most of the current proposals are designed to solve very specific problems, while they could not provide effective generalizations (more information is given in Section 2.1.1).

The approach pursued in [1] instead envisions a DT of the city organized into layers, which cooperate and reconfigure to solve specific problems and support high level decisions. In a modular fashion, new layers representing specific facets of the city (e.g., underground service infrastructure, public mobility service, soil physical-chemical properties) may be added as the functional requirements of the DT change. Among these, a *geometric layer* represents the morphology and physical features (either built or natural) of the city. The geometric layer is represented as a 3D model, encoded as a triangular mesh, where salient semantic entities, like buildings and streets, are evidenced using the mechanism of semantic annotation [7]. The annotated geometric model can reference information related to specific locations (areas, structures, buildings, etc.), needed to describe the urban processes that we want to monitor or predict.

In this paper, we will focus on the process of building the geometric layer of the DT, describing the generation of the geometric 3D model of the city from available data, and the model annotation to achieve a semantic 3D representation of the city which is able to answer specific queries of interest for urban management. Also, two practical examples are demonstrated to query the geometric layer: the computation of points in light/in shadow at a specific date and time, and the automatic extraction of morphology-related information from the 3D model, allowing us to measure dimensions that are nowadays measured only physically.

## 2. Materials and Methods

### 2.1. Background on 3D Modeling in the UI Context

3D models digitally represent shape and features of real objects or phenomena characterised by a three-dimensional nature, i.e., having a spatial extension, which can also bear knowledge in relation to the context of use. In medicine, for example, 3D images are commonly used in diagnostics; in the cultural heritage sector, the digitisation of artistic or archaeological works is increasingly used for conservation, dissemination, and support for documentation and restoration; in the geosciences, 3D models are used for environmental monitoring and risk assessment [8,9], and so on. Furthermore, digital representations enable the application of mathematical models and algorithms to perform analysis, simulation, prediction, that are useful in responding to societal demands and needs.

There are two fundamental approaches to 3D geometric modeling: design and reconstruction from real data acquisition.

In the first case, a designer produces a 3D model from a conceptual idea and develops it using 3D modelling software systems such as Maya [10], 3DStudioMax [11], Catia [12], etc. In the context of construction, the 3D modelling of buildings is deeply linked to the knowledge of the various professionals involved in the design, which is represented and integrated in an information system called "Building Information Modeling" (BIM). BIM [13] is a methodology rather than a simple software, and allows the collaboration of Architecture, Engineering and Construction (AEC) experts in generating a digital model containing data on the entire life cycle of the work, from design to construction and up to its demolition and decommissioning. Also, BIM allows creating a dynamic information model, including modifiable parametric elements, which determine the size and appearance of the building. The same approach can be followed for modelling bigger entities, such as city districts (District Information Modelling—DIM [14]) or even entire cities (City Information Modelling—CIM [15]). Following BIM, buildings are "constructed" by means of a virtual model, which is a "digital twin" at the scale of a single building containing layers on geometry, materials, load-bearing structure, thermal characteristics and energy performance, installations, costs, safety, maintenance.

Besides and beyond BIM, the CityGML standard [16] defines a conceptual model and exchange format for the representation, storage and exchange of virtual 3D city models.

It facilitates the integration of urban geodata for a variety of applications for Smart Cities and Urban DTs, including urban and landscape planning and BIM. In particular, CityGML defines five levels of geometric and semantic detail (LODs) from the lowest (a bounding polygon on the terrain model) to the highest (a 3D building containing indoor features), as depicted in Figure 1; further refinements have also been proposed, e.g., [17]. However, the constructed building might geometrically deviate from the initial design, or change during time; furthermore, in practise, very few city buildings have a BIM counterpart, and those which have are often modeled at a low level of detail.

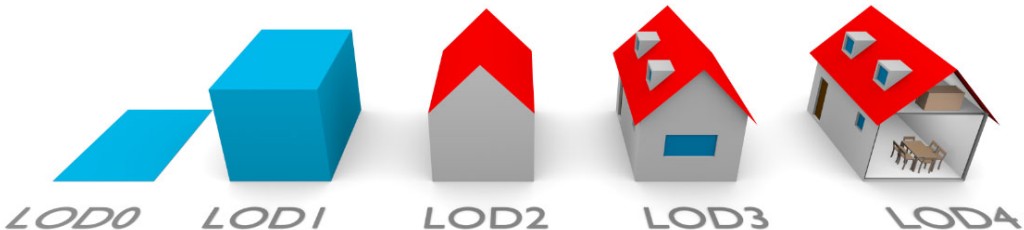

**Figure 1.** The five LODS of buildings in CityGML (image from [17]).

In the second case, a "copy" of a specific real entity is constructed from data measured on the object itself. This type of "measurement" varies depending on the object and the instrument used for this operation, known as *acquisition*. For example, modelling an object means either acquiring a set of points (their coordinates in three-dimensional space) by means of a laser scanner, or taking photographs of the object from every point of view and use photogrammetry techniques to derive the 3D coordinates of the points by correlating their positions in different images. In both cases, the visible surface of the object is acquired and a *point cloud* is obtained, to be further used to reconstruct the surface of the corresponding digital object. Such procedure is called *reconstruction* [18].

Data acquisition at geographical scale tends to be carried out by aerial means, either through images or point sets captured by laser beams (the latter is known as LiDAR - Light Detection and Ranging). The beam signal is fully blocked by built structures, like buildings; however, the most sophisticated LiDAR systems can "go beyond" vegetation and record points on the ground, saving laser first-pulse and last-pulse returns. If there are multiple returns for a single pulse, the first is surely on the highest surface (e.g., the top of a tree) and only the last could be on the ground. So, from the "raw" LiDAR point cloud, two different models of the surveyed area may be reconstructed, called Digital Surface Model (DSM) and Digital Terrain Model (DTM), respectively, the first being the collection of points on top surfaces of buildings, trees, towers, and other features elevated above the bare-earth, and the second being the representation of the bare-earth terrain. The computation of the bare terrain in urban areas requires an intensive processing (outlier removal, point classification, structure recognition, interpolation) and it is not error-free. Finally, DSM and DTM points can be structured as a raster image of equally-spaced z values [19]. Figure 2 shows a LiDAR acquisition, the reconstructed DSM and DTM images, and the mesh obtained by triangulating DSM/DTM points.

While the resolution of the raw LiDAR data refers to the number of sampled points per unit area, the DSM/DTM resolution obtained by processing the LiDAR data is, in general, lower, and corresponds to the cell size of the uniform grid/image. The 3D model resolution must be sufficient to capture the morphology of the territory at the level of detail that will be required by the applications. The higher the resolution, the greater the level of detail at which the model can be observed. If high resolution data is captured, the 3D urban model will have a strong and detailed geometric correspondence with the real city. However, unlike the BIM, the reconstructed model is almost purely geometric and lacks much of the information needed to interpret urban processes; limited semantics might be given by the LiDAR classification of points at acquisition time (e.g., ground, vegetation,

water) [20], if raw acquisition data are available. Otherwise, the geometric model needs to be semantically enriched with knowledge coming from other sources.

In our approach, we start from acquired data to keep a strong morphological correspondence with the built and natural environment as is, and then enrich the reconstructed model with additional knowledge through the process of semantic annotation, in order to support urban monitoring and decision making.

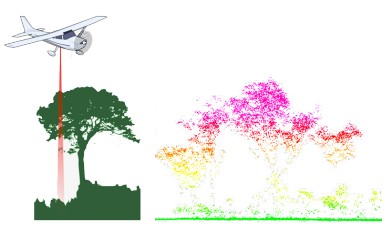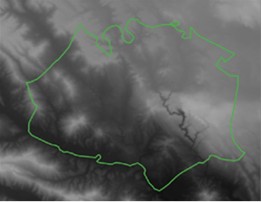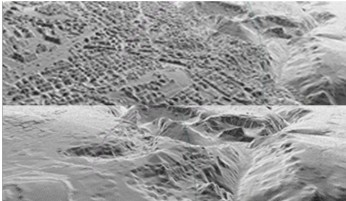

**Figure 2.** From left to right: LiDAR acquisition; generated point cloud; DTM raster image (gray level representing elevation); reconstructed DSM—DTM surfaces.

### 2.1.1. Digital Twins of Urban Contexts

As already introduced in Section 1, several attempts have been made to construct DTs of urban contexts. In [21], a 3D model of the New York City is obtained starting from 2D and 2.5D data and made openly available, but (i) the base of the buildings have been set on zero elevation and (ii) the model has not been integrated with other services of the city, thus limiting the potential of the model to visualization. A similar result is presented in [3,5], where a 3D representation of specific areas of the city is acquired via laser scanning techniques and combined with digital elevation models with the main purpose of visualising and communicating the appearance and extension of buildings to ease urban planning, while actual simulations (e.g., wind/pollution) are not performed on the 3D replica. The authors mention that the introduction of semantized geometries would allow us to go far beyond pure building geometry, even if this result was not obtained in the projects. Our aim is to construct an operational geometric layer, which services of the DT can interrogate and use to achieve more complex goals.

In [4,22], two different 3D models are generated for the city: one "reality mesh model" tightly resembling the real city that is useful for visualization purposes and one "semantic 3D city" with a lower precision but embedded information that is useful for simulation (e.g., wind simulation). With respect to this approach, our method follow a similar path for building generation, but achieves a single geometric model based on real data and augmented with semantic annotations. The final model embeds *queryable* geometric representations (e.g., the street arcs), needed to reach the objectives of urban management stakeholders (see Section 2.1.2).

Finally, in [6] a huge effort has been spent in the generation of a fully queryable 3D model representing the reality, but this has been done with a design approach, thus avoiding automatic reconstruction of the city representation, which is the main focus of this work.

### 2.1.2. State of the Art in City Reconstruction

3D environments can be manually modelled through specialized software such as CityEngine [23,24]. CityEngine is a 3D modelling software that implements a procedural modelling approach to generate urban scenarios importing 2D and 3D data from different map sources. However, the learning curve of the generation rules for the inexperienced user is not negligible. Therefore, the research is focusing on automatic or semi-automatic methods to derive a 3D representation given a 2D map of a city. The most used is Open-StreetMap (OSM) [25] as the most complete open source map dataset. Map elements are described through their 2D geometry footprint, and might be also characterised by optional attributes such as 3D terrain elevation, building floors and height, roof shape and colour.

OSM is built by a community of volunteers, who map or revise their local surroundings, with their own tools and more or less reliably. Therefore, it is not unlikely that a building is described in high detail while the adjacent ones are just roughly outlined; sometimes buildings are roughly defined, with intersecting boundary elements; it also happens that different contributors mapped the same building differently. Therefore, the OSM data need to be cleaned and validated in order to interpret the building footprint correctly and integrated with additional data to derive the necessary attributes to reconstruct the 3D model, the elevation being the most crucial. Ref. [26] derives the height of buildings from the information about floors, considering an average height of 3.2 m per floor; otherwise, it assigns a default height according to the type of building (e.g., assuming an average of three floors for buildings of type "residential" or "hotel"). Unfortunately, many buildings show the same height, due to a general lack of attributes in the OSM description. In [27] the elevation is obtained from an aerial point cloud dataset, e.g., acquired by LiDAR, semantically decomposed/labelled based on the OSM polygons. This approach has the positive feature of expressing the elevation of ground points as well. Additionally, a number of applications exist to produce a 3D from the bidimensional map [28]; among these, OSM2World is an open source project using the elevation attributes or, when unavailable, DEM data from OpenDEM (Open Digital Elevation Model) [29], an open data project for collecting and improving a free digital elevation model of the earth. We followed a similar approach for the geometric model of Matera.

*2.2. The Matera DT*

Matera is a city in the region of Basilicata, in Southern Italy, with a complex morphology with deep ravines and bare highland plateaus, one of the most evocative landscapes of the Mediterranean. The heart of the historical centre is Piazza Vittorio Veneto and the Hypogeum located underneath, built three thousand years ago, consisting of numerous underground levels and containing a huge water reservoir. The adjacent "Sassi district" is a complex of cave dwellings carved into the ancient river canyon. By the late 1800s, the site became noted for intractable poverty and poor sanitation. Renewed vision and investment led to the cave dwellings becoming a noted historic tourism destination and a vibrant arts community, UNESCO world heritage site since 1993. After being European Capital of Culture in 2019, Matera was selected as an innovation hub, to host the "House of Emerging Technologies" funded by the Italian Ministry of Economic Development, including the realisation of the DT of the city.

The DT's goal envisaged in the UI perspective is to provide support to the activities of government, administration and city management. Examples of such activities could be monitoring and prediction of vehicle traffic or air pollution status, and decision-making based on evidence captured by the DT sensor network and by analytic capabilities. The DT is useful also for citizens and associations in the city who can suggest and evaluate the effect of changes on urban space use, by making proposals based on well-documented arguments (participatory urban management).

The reasoning process behind the processes described above are based on a tight interaction between the geometry of the built structures and the processes that take place in it. Indeed, morphological information can be extracted from the 3D model and can contribute, for instance, to assess the accessibility and the perceived comfort of urban spaces and routes: slope magnitude, size and height of steps, availability of ramps for wheelchairs can determine preferences or even accessibility requirements for different user profiles, and these features can be identified in a 3D model of adequate resolution. Moreover, the 3D model allows the computation of points receiving sunlight at a given time and date, with solar radiation being an important comfort factor during summer or winter. Furthermore, a 3D representation could provide the algorithmic setting for the simulation of pollutant in air. A fine detailed 3D model could also offer an informative visualisation of distinctive areas of the city centre and a remote, virtual and immersive visit

experience of sites that are not easily accessible, like the rupestrian churches carved into the limestone in the Matera case study presented in this paper.

One factor that could hinder the potential of high-resolution 3D models is related to the costs related to the 3D acquisition: this is actually one of the motivations of the work presented, which demonstrates how the 3D model construction can start by simply using data available in the public domain, and possibly refined as soon as more data are acquired. Hence, we used available data from a previous national survey: specifically, a DSM and a DTM model, in the form of GeoTiff images, with resolution corresponding to 5 m × 5 m (pixel width and height), provided by the geo-portal of the Basilicata region [30]. In the following, we describe the processing and the achievements obtained with the available data.

### 2.2.1. Reconstruction of the 3D Geometric Layer for the Matera DT

As anticipated, we implemented a similar approach to [27], exploiting 2D information from OpenStreetMap [25] and adding elevation data from external sources. We proceeded by cross-referencing the building and streets (only OSM *ways* and *relations* with tag "building" or "highway" are taken into consideration in the current implementation—this is done by filtering them in QGIS [31]) from OpenStreetMap with the DTM and DSM to derive their real elevation information as follows:

- resolve geometric and topological issues in building boundary polygons;
- insert building boundary polygons and street segments as constraints for the triangulation of the area between buildings(i.e, elements that must be represented as edges in the final mesh). We use Shewchuk's Triangle library to generate the constrained triangulation [32]);
- translate each point in the triangulation so that its elevation corresponds to that of the DTM cell in which the point is originally contained. Points belonging to buildings are set so that their elevation correspond to an average of the DTM cells in which the building is contained;
- assign each building a height equal to the average difference between corresponding DSM and DTM cells containing the building and extrude the building base accordingly;
- close the extruded boundary by triangulating the corresponding polygon (again, using Triangle).

This approach allows generating approximations of the buildings at the LOD1 of the CityGML standard, with buildings as basic shapes without roof and façade details. The main challenge is to fix geometric and topological inconsistencies in the OSM data: indeed, many of the building footprints present topological defects (open polygons, self-intersections, duplicated points, etc.). An example can be seen in Figure 3 (left). To solve the issue, polygons are pre-processed within the QGIS by (i) buffering the polygons by a distance of 1 cm (allows us to fix the *gap* error presented in [33]) and (ii) dissolving the result (allows us to fix the *overlap* error presented in [33]), while reducing the error introduced by different users modelling the same building (inner polygons are merged and only the inner-most profile is kept). The result is then exported using the local coordinate system (WGS 84 UTM 33N EPSG 32633).

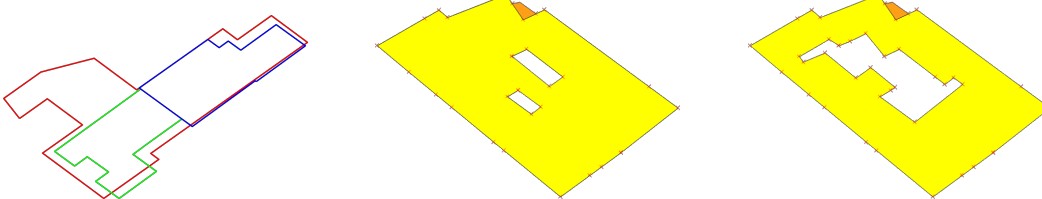

**Figure 3.** Some examples of problematic boundaries in OSM: on the left some intersecting polygons; on the center and on the right, the interior boundary is defined in two different ways into the same OSM *multipolygon relation* defining the building.

As can be seen, the OSM data resolution is sufficient to create a 3D model representing the most modern part of the city, which develops on a fairly flat area and exhibits a certain spacing between the built elements. On the other hand, the generated model is insufficient for the Sassi area, and for an impact visualisation or for the detailed geometric analysis required by some objectives. This can be easily seen in Figure 4. Since, unfortunately, no higher resolution LiDAR data are available for the area, we refined the model as much as possible as follows:

- segments corresponding to roads are refined inserting additional points at each intersection between the segment and the DTM grid: this allows us to add points of known elevation;
- the ground surface is also refined accordingly (see Figure 5).
- the overall terrain mesh is refined applying a refinement method provided by Shewchuk's Triangle library [32].

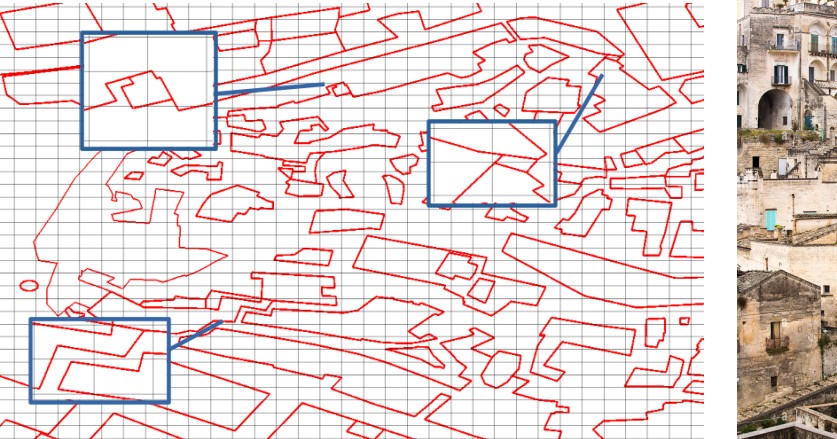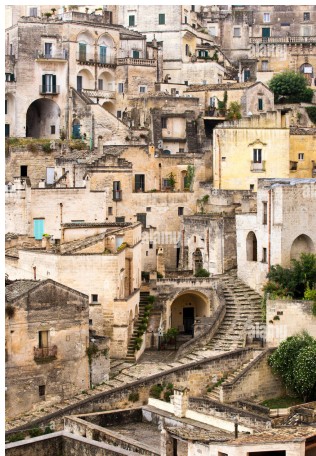

**Figure 4.** On the (**left**): a view of the DTM/DSM grid (corresponding to the pixels of the GeoTIFF file) with impressed building boundaries in the Sassi area. As can be seen, the cell grid size is too large to recognise correctly streets and to diversify close buildings because their points fall within the same pixel. For example, the elevation associated to the small buildings in the top highlighted areas will often be the same of the adjacent ones. The same applies to the streets such as the one in the lower highlighted area, where huge differences in the elevation of the points composing the street may happen if adjacent buildings' bases have substantial differences in elevation. On the (**right**): some alleyways of the Sassi area.

Triangle [32] is the reference method for surface mesh generation. The applied refinement is based on triangle quality criteria, i.e., a threshold is set to the smallest triangle angle (20° in this implementation). This produces a slightly finer model, adding new triangles and new vertices, whose elevation is derived from the DTM cell where they fall into. On the other hand, this method ensures that triangles are well formed without nearly-degenerate cases (angles very close to zero), thus guaranteeing numerical robustness for the downstream processing algorithms. The result is shown in Figure 5 (right). As can be easily seen, the final result still lacks sufficient resolution in the Sassi area. However, this allows us to obtain more pleasant rendering and more reliable values regarding road slopes, feature that is of key importance in the broader context of the city DT (see Section 2.2.2).

Once the geometry of the area has been generated, we semantically annotate the model. So far, we just import elements from OSM, namely buildings and streets. By labelling street edges and building triangles with the corresponding OSM indexes, we create a biunivocal association between portions of the geometry and specific OSM entities and their attributes. Such an information can later be interrogated and analysed in relation to their geometric component.

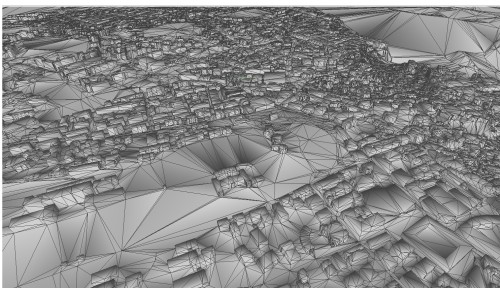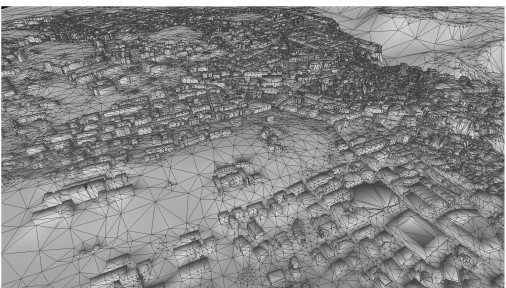

**Figure 5.** Result before and after the refinement of the 3D model.

To annotate buildings (sets of triangles) and streets (sets of edges) we follow the same approach as in [34]: during the 3D model generation process, we automatically create an "annotation" entity for each building, containing the indices of the corresponding triangles; these are obtained with a region growing approach: starting from a triangle inside each polygon (e.g., the left triangle of the first edge of the polygon, since they are consistently ordered counterclockwise by construction). Beside the geometry of the annotated region, the OSM id as well as any additional metadata about the building are added to the annotation. Similarly for street arcs, we create an annotation entity containing the indices of consecutive vertices defining these edges, along with information about the OSM id of the arc and other useful metadata.

Figure 6 shows the refined model, with entities of type "building" and "street" highlighted.

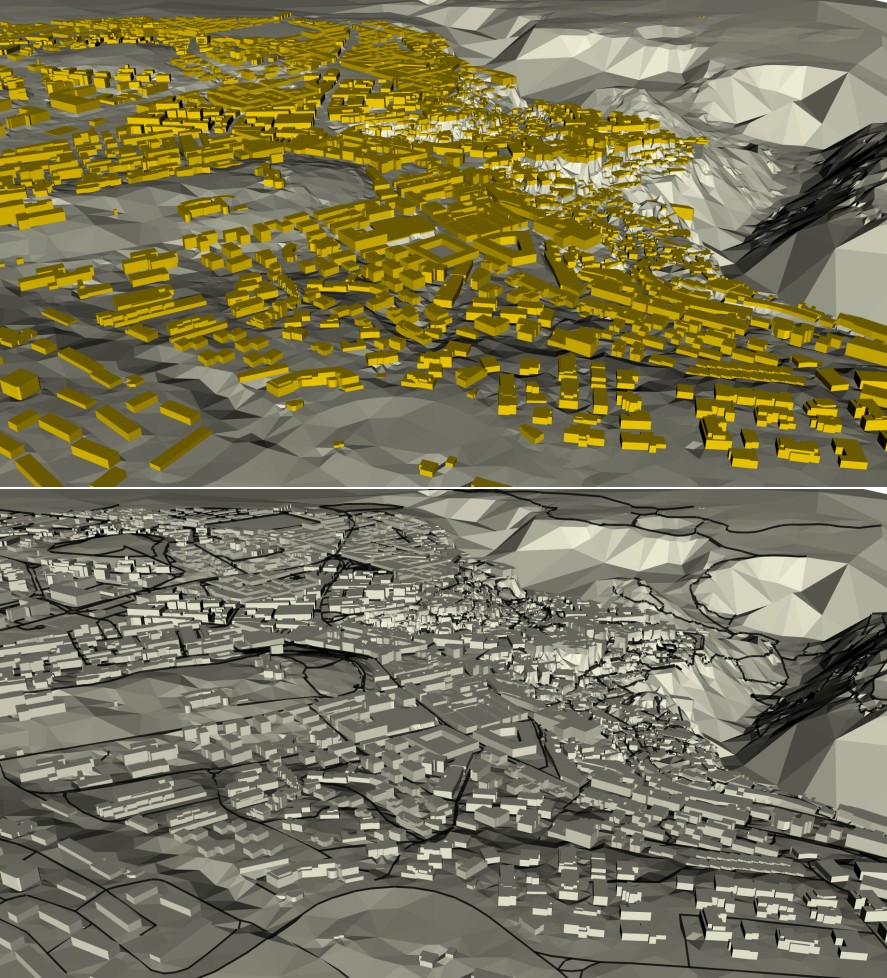

**Figure 6.** From up to down: the refined 3D model with buildings and the road network highlighted, respectively.

### 2.2.2. Morphological Characterization

We designed a morphological characterization service, which performs shape analysis on the 3D city model and annotates geometric elements with the extracted morphological measures/dimensions. So far, the slope analysis of the road network has been implemented. The road network is represented in OSM as a graph, whose nodes are the branches of different routes and whose arcs are road sections connecting two nodes without branches. Each arc has been inserted as a set of edges in the 3D model and annotated consistently.

For each edge annotated as "road", its slope is calculated from the elevation of its incident vertices as the angle formed by the segment and the XY plane. In general, road arcs are composed of a sequence of edges. The total slope of an arc is an aggregated value of the individual segment slopes. The service provides several alternatives, namely minimum, maximum and average slope.

Beside annotating edges with their slope value, the results of the morphological characterisation of the routes are saved in an output file (specifically, a .json file storing the road graph with minimum, maximum and average slope for every arc) that is shared among the DT services.

### 2.2.3. Sun/Shade Computation

The 3D model is indispensable for a detailed calculation of the shaded/illuminated areas at a given time of day and time of year, taking into account the built structures that can shadow ground points.

The algorithm is based on the calculation of the intersections of a ray of known inclination with the city model. All the vertices in the model are tested, and currently the result is binary (in light/in shadow). The service requests as input the year, day and time, used to compute the angle of the sun's rays.
The implemented algorithm proceeds as follows:

- The angle of the sun's rays is computed, via the SOLPOS library [35], on the basis of the input parameters (date and time);
- An intersection check is made between the ray and each vertex in the 3D model: if there are no intersections, then the vertex is illuminated; otherwise, the vertex is shaded.

## 3. Results

Figure 7 shows some results of the slope characterization of the road networks, using local, per segment values, while Figure 8 shows the average value on arcs. Future developments of this service may include the analysis of the width of streets, the slope and width of roofs, the detection and quantitative analysis of structures such as pavements, ramps, stairs, the identification of rough paths, etc. Morphological analysis is based on the 3D geometric model and exploits the information contained therein. Thus, the accuracy of the characterisation results is strongly linked to the quality of the model, i.e., the resolution of the input data. For example, the detection of a pavement or a ramp requires a level of detail in the order of 5 cm in height. The low resolution however affects the slope characterisation as well, at least in the Sassi area (Figure 7, middle and right). Here the reconstruction, and consequently the representation of the routes and the calculation of their slope, are visibly coarse. A critical point is therefore the availability of data for a more refined 3D model, and consequently, more precise and reliable measurements.

An example of the shading calculation is shown in Figure 9. This service can be used both for monitoring the current shading and for visualising past or future situations. The calculation time depends on the complexity of the 3D model, i.e., the number of vertices to be characterized and the number of surfaces to check for potential intersections. The output is a shared shadow map, actually a bit vector storing at the $i$-th position 1 if the $i$-th vertex is illuminated, 0 otherwise. At the current state, the output is limited to potential illumination/shading based on date and time: no real time information or weather forecasts on sky coverage are taken into account; similarly, the output is binary (0 = shadow, 1 = light)

while it could be a real value in the range [0–1] if sky coverage from satellite data or forecasts were taken into account, and/or physical light behaviour is considered (e.g., different shade gradations given by reflected rays). As with the morphological characterisation service, the quality of the shading/radiance estimate depends on the resolution of the 3D model. In particular, since buildings have been extruded from the plane, they are seen as closed blocks; situations like porches or other partially covered areas are not currently represented, nor taken into account for the shading calculation.

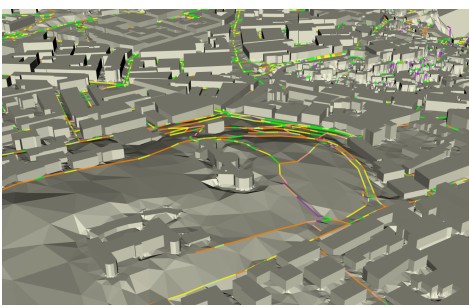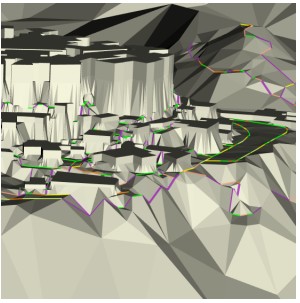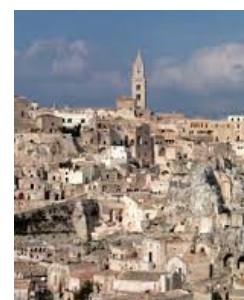

**Figure 7.** The road network is depicted according to local, per segment slope (from green = null slope to red = maximum slope −30° in this implementation; values above this threshold are visualised in violet for suggesting unrealistic values). Note that, in the Sassi district (**middle**), the slope computation is much more noisy than in the modern area (**left**), as it suffer from the low resolution with respect to the complex morphology of the area degrading to the river (**right**).

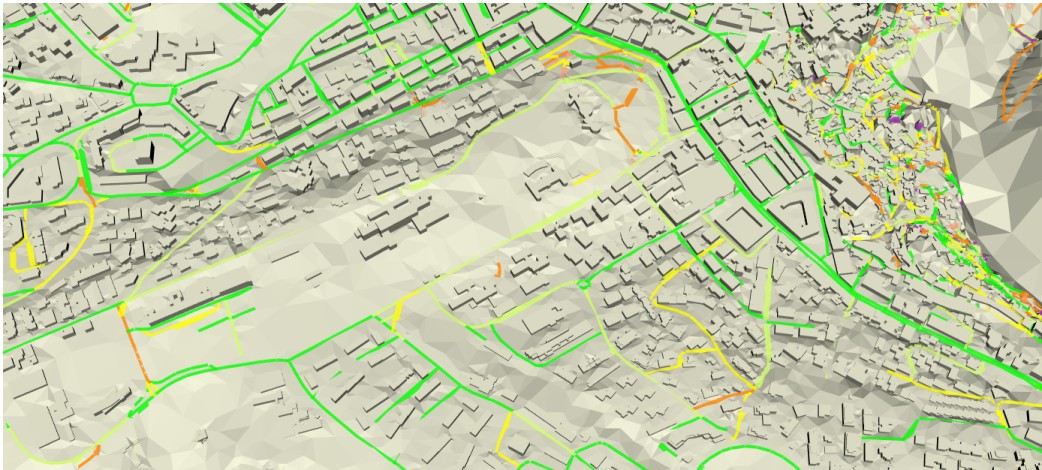

**Figure 8.** The road network is depicted according to the average slope (from green = null slope to red = 30°). Here the representation is smoother since an arc (between two branches) has an aggregated slope value.

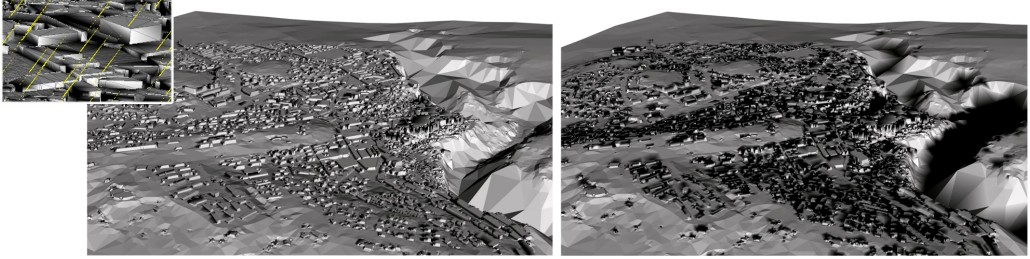

**Figure 9.** Solar illumination expected on 15 August (**left**) and 25 December (**right**) at the same time (9:00 a.m.). In the detail, some incident rays used in the computation.

The two services, beside giving a direct answer to a user query, such as "what is the slope of this route?" or "is the playground still under the sun?", are designed to provide

information related to the morphology of the city to further services and realise more complex objectives thank to their integration. For instance, both the route slope and the sun/shadow characterization are likely to impact the sense of well-being perceived by a pedestrian walking in the city (plus other environmental factors like noise, traffic, air pollution, etc.). A path planning algorithm could help visitors optimising their visit [36], taking into account the two characterizations as arc or node weights and determining an optimal route according to the user preferences, e.g., shortest path versus minimum slope versus as much shade as possible. The slope analysis is also functional to determine accessible paths for impaired users: for instance, a steep street is not accessible for a hand-propelled wheelchair. Finding an accessible path for a wheelchair user corresponds to find a route where each segment of arc has a slope less or equal to a certain threshold (as a reference, standard wheelchair ramps exhibit about 5 degrees of incline).

Finally, the annotated 3D model can effectively render and communicate geo-referenced information from other layers, such as data streams from real-time sensors. In Figure 10 we imagined to visualise different points of interest (e.g., museums, historical buildings) according to an occupancy level, where the number of people can either correspond to the actual occupancy communicated to the DT by webcams/sensors, or be predicted by data-driven models [37]. At higher level, Artificial Intelligence (AI)-based models will provide the Digital Twin with 'intelligent' capabilities [38], for instance, to abstract and aggregate information generated by different layers, to organise sequences of interventions on the territory, but also to react autonomously to certain situations that may dynamically emerge in the urban context (e.g., give an alert if the occupancy level of a point of interest is too high).

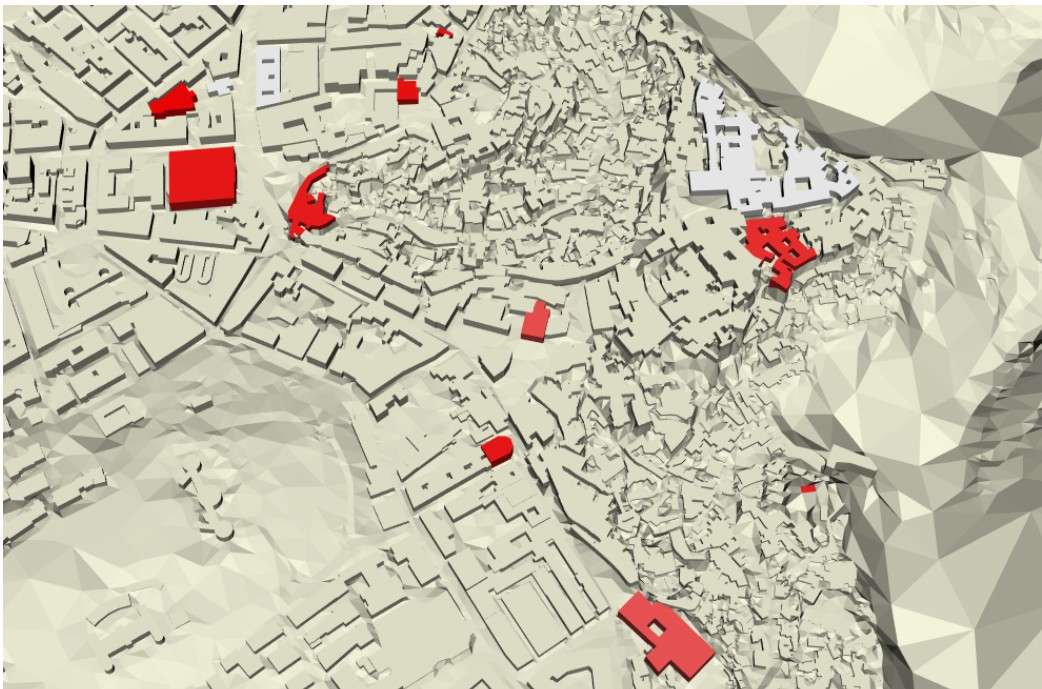

**Figure 10.** Example of visualization of further geo-referenced information from real-time sensors: from white to red, increasing occupancy rate of Points of Interest.

## 4. Conclusions

In this paper, we proposed an approach for quickly designing and developing the geometric layer of a urban digital twin, describing the morphology and physical structures of the city, together with additional knowledge related to specific locations. This method exploits both data coming from OpenStreetMap and land morphology obtained with LiDAR techniques, namely Digital Terrain Models and Digital Surface Models. Furthermore, the geometric layer is enriched by semantic information annotating geometric portions.

The potential of this approach is shown by generating the geometric layer for the digital twin of the Italian city of Matera and encoding the information regarding buildings and streets. it is used to answer specific objectives posed by the Municipality: to extract the in light/shadow areas given date and time, to automatically extract the morphology-related information about streets and to communicate the level of occupancy of certain Points Of Interest. The focus of our work is not in achieving a top quality 3D model, but rather in constructing a functional geometric model of the city, exploiting at best the available data, trying to achieve a representation that is able to support the geometry related queries. In the future, we expect to integrate this with several other layers to provide answers to complex queries: for example, adding a physical-chemical layer would allow integrating slope information with soil properties information at a specific location, potentially highlighting possibilities of soil saturation, water stagnation, runoff , etc.

In the next phase of the project, we are going to include further salient entities for the project objectives, such as squares, green areas, public and commercial services, urban furniture and facilities, and others. Moreover, we are currently developing additional integrated services that use the geometric model: for example, the path planning using slope and sun/shadow as preferences; we are also integrating historical/forecast satellite data to compute the actual solar radiation according to the sky conditions and estimate an average radiation per year. Future development will focus on the monitoring and simulation of pollutant diffusion in the air, computed on the 3D model of the air volume.

Finally, another hint for future works is given by the challenge of generating 3D models at higher CityGML LOD; for example, LOD2 models present features related to roof shape and façade details. These elements can define more reliably the sun ray incidence on the perceived temperature and the total radiation and the thermal gain of solar implants.

**Supplementary Materials:** The data presented in this study are available online at https://www.mdpi.com/article/10.3390/ijgi11060343/s1.

**Author Contributions:** Conceptualization: Andreas Scalas, Daniela Cabiddu, Michela Mortara; Formal analysis: Andreas Scalas, Daniela Cabiddu; Funding acquisition: Michela Mortara, Michela Spagnuolo; Investigation: Andreas Scalas, Daniela Cabiddu; Methodology: Andreas Scalas, Daniela Cabiddu, Michela Mortara; Project administration: Michela Mortara, Michela Spagnuolo; Software: Andreas Scalas, Daniela Cabiddu; Supervision: Michela Mortara, Michela Spagnuolo; Validation: Andreas Scalas; Visualization: Andreas Scalas; Writing—original draft: Michela Mortara, Andreas Scalas, Daniela Cabiddu. All authors have read and agreed to the published version of the manuscript.

**Funding:** The CTEMT Project is financed by the Ministry of Economic Development (Italy) with convention prot.G.0010812/2020 - U - 05/02/2020 signed between the MISE and the Municipality of Matera. This research is part of CTEMT Project and in particular it is placed within work package 1 "Realisation of the Urban Digital Twin" developed by the National Research Council (CNR-Italy) with the technical-scientific support of the National Institute of Urban Planning (INU-Italy) signed with convention prot_65562_06102021.

**Institutional Review Board Statement:** Not applicable.

**Informed Consent Statement:** Not applicable.

**Data Availability Statement:** The data presented in this study are available in the Supplementary Material.

**Acknowledgments:** The authors would like to thank the municipality of Matera and the Italian National Research Council for their support and in particular the involved teams from INM, IASI, ISTC, ICAR, IEIIT and in general from DIITET.

**Conflicts of Interest:** The authors declare no conflict of interest.

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
