# Peer review of "Potential of the Geometric Layer in Urban Digital Twins"

_ijgi, doi:10.3390/ijgi11060343_

Round 1

Reviewer 1 Report

The work proposed by the authors presents a strategy to obtain 3D models of outdoor environments such as urban spaces. These models are focused on providing an heterogeneous semantic information container to support advanced services such as city management or services to improve the quality of life of citizens. To this end, they propose to integrate data from an aerial survey with the information available on the OpenStreetMap platform. Finally, they propose two services that exploit the resulting model to provide information about street slopes and solar incidence.
Although the solution proposed by the authors is interesting and represents an advance of the state of the art in the subject treated, there are minor aspects that in my opinion should be addressed before publication. For example:
  1. A little more detail is missing in the paragraphs from line 208 to 217. For example, it is mentioned that polygons are pre-processed with QGIS software to solve OpenStreetMap inconsistency problems. How are they solved? How is it solved when the same building is modeled differently by two different users? This last problem is shown in Figure 3 (center and right), but is not referenced or addressed throughout the text.
  2. Line 215 mentions that the model is insufficient for the Sassi area. However, no further details are given and the reasons for this inadequacy are not developed. Perhaps a brief explanation supported with images in this part of the text would be interesting.
  3. How does the proposed in 218-221 help to refine the lack of higher resolution LiDAR information of the Sassi area?
  Although it is not essential, it might be interesting:
  1. Make the proposed model of the city of Matera publicly available to the reader.
  2. Add some detail of the calculation times mentioned in line 283. Perhaps the times with a given hardware configuration.
  Typo corrections
  1. In line 210 reference is made to Figure 4, but I believe it is Figure 3.
  2. A parenthesis is missing in line 221.
  3. The quality of the right subfigure of Figure 6 is improvable.

Author Response

The work proposed by the authors presents a strategy to obtain 3D models of outdoor environments such as urban spaces. These models are focused on providing an heterogeneous semantic information container to support advanced services such as city management or services to improve the quality of life of citizens. To this end, they propose to integrate data from an aerial survey with the information available on the OpenStreetMap platform. Finally, they propose two services that exploit the resulting model to provide information about street slopes and solar incidence.

Although the solution proposed by the authors is interesting and represents an advance of the state of the art in the subject treated, there are minor aspects that in my opinion should be addressed before publication. For example:

  1. A little more detail is missing in the paragraphs from line 208 to 217. For example, it is mentioned that polygons are pre-processed with QGIS software to solve OpenStreetMap inconsistency problems. How are they solved? How is it solved when the same building is modeled differently by two different users? This last problem is shown in Figure 3 (center and right), but is not referenced or addressed throughout the text.

We added the requested explanations in section 2.2.1, lines 282-290.

  1. Line 215 mentions that the model is insufficient for the Sassi area. However, no further details are given and the reasons for this inadequacy are not developed. Perhaps a brief explanation supported with images in this part of the text would be interesting.

The issue is that narrow paths and small buildings in the old city are not captured by the elevation grid, which provides quoted points every 5 meters. We tried to better explain this issue with a new Figure 4 referred to in section 2.2.1.

  1. How does the proposed in 218-221 help to refine the lack of higher resolution LiDAR information of the Sassi area?

Actually, it doesn’t. The refinement is made on the street arcs, which represent segments of the real street with variable length (even hundreds of meters), thus effectively making it impossible to accurately calculate the slope. We tried to reduce the issue as much as possible by exploiting available DTM/DSM data, inserting further points with known elevation that split street arcs in smaller segments. Such a refinement provides a more pleasant rendering and can smooth the visualization and the computation of heights and slopes of roads and streets, but, unfortunately, it is not sufficient to improve the representation of the morphology of the Sassi area. We highlighted this in the text.

  Although it is not essential, it might be interesting:

  1. Make the proposed model of the city of Matera publicly available to the reader.

We are considering it, as soon as it will be finalized, if it is allowed by the project policy.

  1. Add some detail of the calculation times mentioned in line 283. Perhaps the times with a given hardware configuration.

We are currently fixing and making the code more efficient, therefore we prefer to add this information in the final version of the paper, if accepted.

  Typo corrections

  1. In line 210 reference is made to Figure 4, but I believe it is Figure 3.
  2. A parenthesis is missing in line 221.
  3. The quality of the right subfigure of Figure 6 is improvable.

We applied the suggested corrections and improved the quality of the right subfigure. Fixed typos.

Reviewer 2 Report

This paper presents a low cost scenario for creating a 3D model for Digital Twin model based on OSM data and and a DSM/DTM.

Please complete Affiliation information 

The term "geometric layer" that appears in the title and in the text multiple times is not a well known term and it confuses the reader. It should be replaced by another text such as "low cost 3D model" or other.

Line 1: physical and technical assets

Line 41: physical structures are only natural not built / by "knowledge" you mean thematic information / descriptive attributes?

Line 43: again the use of "knowledge" word is problematic, "knowledge and data" does not make sense

Line 44: knowledge technologies ???

 A few remarks on figures:

Figure 3a is not visible. OSM does not keep polygons provided by different users about the same object. Overlapping polygons of this size are due to different objects such as building in a park.

Figure 6a and 6b the road network is not visible

Figure 4: Use the same zoom level and observer position in both pictures.

Bibliography misses information such as year and needs to follow a specific style. Especially on line references need to be completed.

Good luck in publishing your paper.

Author Response

This paper presents a low cost scenario for creating a 3D model for Digital Twin model based on OSM data and and a DSM/DTM.

Please complete Affiliation information 

Affiliation information has been completed.

The term "geometric layer" that appears in the title and in the text multiple times is not a well known term and it confuses the reader. It should be replaced by another text such as "low cost 3D model" or other.

We added a few words about the possible “layers” of a DT (lines 42-44) and tried to better define what we mean by geometric layer: “a geometric layer represents the morphology and physical features (either built or natural) of the city.The geometric layer is represented as a 3D model, encoded as a triangular mesh, where salient semantic entities, like buildings and streets, are evidenced \added{using the mechanism of semantic annotation” (lines 44-50)

Line 1: physical and technical assets

Line 41: physical structures are only natural not built / by "knowledge" you mean thematic information / descriptive attributes?

Maybe there was a misunderstanding: here we used the term “physical” as a synonym of “real” in contrast to “digital”, including both natural and built. We replaced “physical” with “real” in the paper.

Line 43: again the use of "knowledge" word is problematic, "knowledge and data" does not make sense

Line 44: knowledge technologies ???

With “knowledge” we mean all kinds of additional information related to a portion of geometry, including thematic knowledge and descriptive attributes, which can be associated with data in a machine-readable way. Knowledge and data make sense as “data about data”, that is, metadata. The slope of a street can be added as metadata  to the segment representing the street (geometric data); this link is established by annotation.

Knowledge Technologies are the technologies of the Semantic Web (e.g., ontologies): technologies to encode knowledge in a machine-readable way, to share it, to associate it to digital content and to make reasoning on it. The topic and the terms are probably unfamiliar for the readers of this journal. 

However, we would not insert a background section on knowledge technologies, because this is not the focus of the paper. We tried to simplify the text e.g., using “semantics” instead of knowledge where possible.

 A few remarks on figures:

Figure 3a is not visible. OSM does not keep polygons provided by different users about the same object. Overlapping polygons of this size are due to different objects such as building in a park.

We improved figure 3a by focusing on intersection of buildings’ footprints and thickening the boundaries of the polygons. About the overlapping polygon issue, the reviewer is right: what we meant is that interior polygons may be defined in different ways in cases of multipolygons, such as the one reported in Figure 4. We corrected the sentence.

Figure 6a and 6b the road network is not visible

We improved the visibility of the road network in figure 6 by thickening the corresponding segments.

Figure 4: Use the same zoom level and observer position in both pictures.

Fixed.

Bibliography misses information such as year and needs to follow a specific style. Especially on line references need to be completed.

Fixed.

Reviewer 3 Report

The paper is well constructed but it is 'old'. The environmental analysis today makes use of terrestrial and drone data, acquired with spectral and thermographic cameras. The use of satellite images alone does not allow for high accuracy on the urban scale. The 3D reconstruction of the city is only geometric, lacking the physical-chemical characteristics of the surfaces. An overly simplified 'small cube' model, certainly not a webgis. It is advisable to see the state of the art (the references are insufficient), to use more appropriate urban 3D survey techniques and more suitable 3D modeling software. The paper, at present, cannot be accepted.

Author Response

The paper is well constructed but it is 'old'. The environmental analysis today makes use of terrestrial and drone data, acquired with spectral and thermographic cameras. The use of satellite images alone does not allow for high accuracy on the urban scale. The 3D reconstruction of the city is only geometric, lacking the physical-chemical characteristics of the surfaces. An overly simplified 'small cube' model, certainly not a webgis. It is advisable to see the state of the art (the references are insufficient), to use more appropriate urban 3D survey techniques and more suitable 3D modeling software. The paper, at present, cannot be accepted.

We probably failed in explaining clearly the context and constraints of our application. The reviewer is absolutely right about the current acquisition methodologies able to provide much better starting data for the reconstruction, in terms of range of information (e.g., chemical properties of materials), resolution, accuracy. 

Unfortunately, the project did not foresee a preliminary acquisition phase; therefore, the low resolution DSM and DTM were the only available data for the reconstruction. 

The focus of our work is not in achieving a top quality 3D model, but rather, in constructing a functional geometric model of the city, exploiting at best the available data, trying to achieve a representation that is able to support the geometry related queries. In the case study of Matera, results show that the available data resolution is partially sufficient, as the severe complexity of the Sassi district does not allow reliable geometric estimation in that area. We have significantly reworked sections 2.1 and 2.2 to clarify this aspect.

Concerning the lack of other characteristics and being only geometric, this is absolutely true. Indeed our purpose was to create the geometric layer of the city morphology, to be integrated with other layers to provide answers to complex queries. For example, adding a physical-chemical layer would allow integrating slope information with soil properties information at a specific location, maybe highlighting possibilities of soil saturation, water stagnation, runoff.

Reviewer 4 Report

The article describes a methodology for the implementation of an urban Digital Twin using Open Street Map data (building shapes and roads) and available Digital Surface and Terrain Models to obtain elevation information. In my opinion the topic is interesting but the manuscript, in its current form, has some flaws and issues.

  1. Introduction and state of the art

Introduction should be carefully reworked to critically present the current state-of-the-art on urban Digital Twin and 3D city models. The cited examples (refs. 2-4) are just briefly listed and should be better described to make the reader able to understand the state of the art and appreciate the new contribution provided by the authors.

In addition, some conceptual flaws can be pointed out:

  • Lines 57-60: “bearing knowledge in relation to the context of use” and “thanks to the informative content they provide”. 3D models do not always provide information. For instance, a mesh model gives information on the shape of the object, on its geometry, but may not be semantically enriched.
  • Section 2.1. I have some confusion about whether to distinguish between design and reconstruction approaches. Since the article focuses on creating a digital twin of a real city, it is clear that a reconstruction approach is considered. Then the point of interest should be on data acquisition, processing, and semantic enrichment. As it is written, only the design approach seems to provide an information 3D model, while the reconstruction approach seems to correspond to a 3D survey.  This is not the case and, in contrast to what is said in lines 115-118, the two approaches do not need to be combined. In both cases an information model can be obtained: in the former approach, the digital model is made from a design idea, in the latter from survey data. Only the starting point changes and, when working on existing buildings or cities, a preliminary survey is required.
  • Line 98: “Such procedure is called reconstruction”. Please provide some reference because, for my best knowledge, reconstruction is not a standard definition.
  • Lines 68-83: BIM is not only part of the design approach. A Building Information Modelling can also be realized for existing buildings (search “As Built BIM”, “Scan to BIM methodology”, “Historic BIM”). In addition, since the article refers to the urban scale it might be more appropriate to refer to information systems in general, including BIM at the building scale, DIM (District Information Modeling) and CIM (City Information Modeling) at the urban scale.
  • References to BIM and CityGML should be added
  • Lines 66-70: please add some reference also to BIM authoring software.
  • Lines 76 : “The building is “constructed” before its physical realisation”. Not always. See the comment to lines 68-83 about BIM for existing buildings.
  • Lines 102-106: For a better comprehension, the definition of DTM and DSM should be modified and improved, assuming not all the readers are knowledgeable in this field.

The difference cannot be explained by relying only on laser beam pulses. The use of first and last pulses to determine DSM and DTM only applies to vegetation. In the case of the built environment (buildings, bridges, etc.) the DTM is obtained differently, since the laser beam does not pass through the buildings. Furthermore, both DTM and DSM can be obtained via photogrammetric surveys that do not involve pulses.

  1. Methodology and results

The methodology presented in the paper is not particularly innovative and applies the same procedure as in [12] to a different context. The paper presents initial results which, according to the authors, need to be further investigated and implemented. The results show that the digital model essentially fulfils (with limitations) two tasks: shadow and slope calculation. This contrasts with the initial statement of general purpose (lines 37-40) to cope with the limitations of the previous examples which “are designed to solve very specific problems (line37)”.

Some remarks:

  • Line 167: How are historical data integrated into the model?
  • The results should be validated, in particular those related to slope. How reliable is the analysis presented?
  • Why is the 3D important? Many of the presented goals can be achieved by using GIS analysis
  • Line 181: An higher level of detail should be required to allow virtual and immersive visit. LOD 1 is not sufficient.
  • Lines 230-236: which procedure has been implemented for semantic annotation?

  1. Refences

More references should be added. Links to web pages are missing in some cases (e.g. Refs 2, 4, 14, 17).

Author Response

The article describes a methodology for the implementation of an urban Digital Twin using Open Street Map data (building shapes and roads) and available Digital Surface and Terrain Models to obtain elevation information. In my opinion the topic is interesting but the manuscript, in its current form, has some flaws and issues.

  1. Introduction and state of the art

Introduction should be carefully reworked to critically present the current state-of-the-art on urban Digital Twin and 3D city models. The cited examples (refs. 2-4) are just briefly listed and should be better described to make the reader able to understand the state of the art and appreciate the new contribution provided by the authors.

We added section 2.1.1 devoted to describing existing urban DTs and clarifying our contribution.

In addition, some conceptual flaws can be pointed out:

  • Lines 57-60: “bearing knowledge in relation to the context of use” and “thanks to the informative content they provide”. 3D models do not always provide information. For instance, a mesh model gives information on the shape of the object, on its geometry, but may not be semantically enriched.

We have simplified the text, because we thought it was too specific. However, our point is that a geometric model representing the shape of an object may carry information in its context of use even if it is not annotated. The analysis of the geometry can reveal knowledge that was implicitly carried by the shape itself. An example is the shape of bones in the monitoring of bone degradation with time in patients with osteoarthritis. By estimating surface curvature and by comparing volumes locally, geometric shape analysis algorithms can located eroded regions and estimate the evolution of the disease in the patient. Of course, once extracted, this new knowledge can be used to annotate the model and make this knowledge readily available to others.

  • Section 2.1. I have some confusion about whether to distinguish between design and reconstruction approaches. Since the article focuses on creating a digital twin of a real city, it is clear that a reconstruction approach is considered. Then the point of interest should be on data acquisition, processing, and semantic enrichment. 

Digital twins of cities are often created by design rather than reconstruction (for example, in the Virtual Singapore project the Dassault Sistémes 3DExperience platform is used to model the city), of course following data obtained through an acquisition campaign but without an automatic reconstruction method. 

Concerning the focus, the previous text was not clear. In this revision, we tried to point out that, even if we go for a reconstruction process, unfortunately we could not afford an ad-hoc acquisition, but rather we had to use already available data: pre-processed DSM and DTM grids at 5x5 meters, a quite low resolution. Therefore, the focus is on how to exploit at best the available data (processing and annotation), and not on the acquisition. We explain this aspect better in section 2.2.

  • As it is written, only the design approach seems to provide an information 3D model, while the reconstruction approach seems to correspond to a 3D survey.  This is not the case and, in contrast to what is said in lines 115-118, the two approaches do not need to be combined. In both cases an information model can be obtained: in the former approach, the digital model is made from a design idea, in the latter from survey data. Only the starting point changes and, when working on existing buildings or cities, a preliminary survey is required.

We removed the sentence about “combining the two approaches”.

  • Line 98: “Such procedure is called reconstruction”. Please provide some reference because, for my best knowledge, reconstruction is not a standard definition.

We added the reference to a recent state of the art on surface reconstruction from point clouds.

  • Lines 68-83: BIM is not only part of the design approach. A Building Information Modelling can also be realized for existing buildings (search “As Built BIM”, “Scan to BIM methodology”, “Historic BIM”). In addition, since the article refers to the urban scale it might be more appropriate to refer to information systems in general, including BIM at the building scale, DIM (District Information Modeling) and CIM (City Information Modeling) at the urban scale.
  • References to BIM and CityGML should be added
  • Lines 66-70: please add some reference also to BIM authoring software.

We now mention DIM and CIM, and we inserted the requested references.

  • Lines 76 : “The building is “constructed” before its physical realisation”. Not always. See the comment to lines 68-83 about BIM for existing buildings.

The reviewer is right, we fixed the sentence.

  • Lines 102-106: For a better comprehension, the definition of DTM and DSM should be modified and improved, assuming not all the readers are knowledgeable in this field.

The difference cannot be explained by relying only on laser beam pulses. The use of first and last pulses to determine DSM and DTM only applies to vegetation. In the case of the built environment (buildings, bridges, etc.) the DTM is obtained differently, since the laser beam does not pass through the buildings. Furthermore, both DTM and DSM can be obtained via photogrammetric surveys that do not involve pulses.

We pointed out more clearly that the laser beam can pass through vegetation only (line 116). In this paper, we don’t focus on the acquisition phase, we simply focus on the processing of data that were already available; in our case, such data were a DSM and DTM geotiff files, preprocessed by the italian administration of Regione Basilicata using LiDAR data from a previous campaign at national level. We simply wanted to provide some background about how these surveys are carried out. We added to the text the observation that LiDAR are blocked by built structures and specified that the difference between first and last pulse is used in this case only. We also better detailed  the difference between LiDAR, DSM and DTM (lines 118-132).

  1. Methodology and results

The methodology presented in the paper is not particularly innovative and applies the same procedure as in [12] to a different context. The paper presents initial results which, according to the authors, need to be further investigated and implemented. The results show that the digital model essentially fulfils (with limitations) two tasks: shadow and slope calculation. This contrasts with the initial statement of general purpose (lines 37-40) to cope with the limitations of the previous examples which “are designed to solve very specific problems (line37)”.

Here we meant that there is an a-priori design intent in the definition of a DT. The reported examples (now better explained in section 2.1.1) take design choices in order to fulfill a specific objective. In our case, the objectives are broad and in our design intent, based on layers, the DT is modular in order to cope, in the future, with new objectives. In this paper, we focus on the geometric layer and aim to demonstrate the importance of the geometric information to answer some queries related to the city. Moreover, we present preliminary results, that are just two examples to support the importance of the geometric layer; we added at lines 435-440 the next services using the geometric representation that we are going to develop.  

Some remarks:

  • Line 167: How are historical data integrated into the model?

Here the discourse is at a general level, and not specific for the 3D model. However, the DT has a storage unit, the Data Lake, where various data, described by metadata, are maintained and shared among services, including real-time data from sensors. Available historical data (e.g., traffic, occupancy, pollution, rainfall) can be stored as well, and accessed, for instance, to train a service to predict visitor occupancy in public spaces. This service has been developed by other project partners, and it is not described here, since it does not use the geometric model.

  • The results should be validated, in particular those related to slope. How reliable is the analysis presented?

The slope accuracy simply depends on the accuracy of the elevation of the extremal point of a segment for which we provide the elevation gap and the overall inclination; of course nothing prevents the real counterpart of the segment to present strong slope oscillations in-between, so there is no guarantee about the real slope. However, the reliability of the estimate also depends on the sampling density of elevation points: from the DTM data, we have quoted points  every five meters. The elevation of these points is reliable: it is certified by the Italian ministry of environment, which acquired and provided the data. Intuitively, this density is actually enough to estimate the average slope of a road: in a five meters range, it is possible to have an abrupt inclination change, but rather unexpected to have strong oscillations between the two quoted points. For example, having the extreme points at the same height and having 1 meter height difference midway is quite unlikely. For sure, our prediction is more reliable than OSM, which uses less dense quoted points. However, the reviewer is right, we did not perform a quantitative evaluation of the slope reliability yet. Moreover, as we mention in the paper, the slope estimation is not reliable in the Sassi area, where the majority of paths are not accessible to vehicles but rather rocky pathways and stairs, where the sampling density of five meters is not even enough to represent the paths themselves. 

  • Why is the 3D important? Many of the presented goals can be achieved by using GIS analysis

We agree with the reviewer that for many cases, a GIS would be not only sufficient but more efficient. However, for specific cases, we argue that the geometric 3D model is crucial. Is GIS analysis able to compute the exact shadow at a specific time/date? We would say no, since geometric computations are involved to evaluate ray intersections with solids representing buildings. We understand that we provide so far only two examples of services using the geometric model, and unfortunately the resolution is limited; our goal is to show the potential of the approach.  

Line 181: An higher level of detail should be required to allow virtual and immersive visit. LOD 1 is not sufficient.

Of course. In the future, we envisage the 3D model to be locally enriched with high-resolution data, coming from additional acquisition surveys that will be carried out to scan some specific points of interest.

  • Lines 230-236: which procedure has been implemented for semantic annotation?

    Details on this aspect and reference to our previous publication describing the procedure have been added at the end of section 2.2.1, lines 322-331.

 Refences

More references should be added. Links to web pages are missing in some cases (e.g. Refs 2, 4, 14, 17).

Fixed. 

Round 2

Reviewer 3 Report

The authors have done very good work. They made the best use of the data at their disposal. The limitation of the work lies in the use of ancient technologies and methodologies. The work would have been awesome five years ago, but now it's old and outdated.

I'll let the editor decide

Author Response

Concerning the city 3D model per se, we agree that new technologies are able to provide much better (detailed) results. However, this work focuses on the potential of quickly generating and using a 3D model imbued with semantics to answer geometry-related queries in the wider context of a city digital twin. The project is ongoing and results in this paper are preliminary. However, we showed the added value of the 3D model with two developed services and sketch the expected future achievements with new services and their integration to support the city management.
We are negotiating with the local administration the possibility of an ad-hoc acquisition in the Sassi area using airplane or drones coupled with terrestrial scanning, and we hope to show a significant improvement in future works.

Reviewer 4 Report

The authors have significantly improved the paper by extending and reviewing some parts of the text and by fixing previous issues. After the revision, the article now warrants publication.

I have just a minor remark, which is probably due to the presence of comments and revisions in the file: Check line 436 because the text is incomplete.

I suggest minor revision to allow the authors to the remark I have indicated

Author Response

We want to thank the reviewer for her/his time and suggestions. We updated the paper with the required correction (there was some issue with the compilation of the latex project).